## PERSPECTIVE

### Remote contractions to mitigate reduced persistent inward current magnitudes in motoneurons of older adults

Jakob Škarabot [ID]

*School of Sport, Exercise and Health Sciences, Loughborough University, Loughborough, UK*

Email: J.Skarabot@lboro.ac.uk

Handling Editors: Richard Carson & Mathew Piasecki

See Original Article here

The peer review history is available in the Supporting Information section of this article (https://doi.org/10.1113/JP283964#support-information-section).

Motor units (MUs; i.e. the $\alpha$-motoneuron and all the muscle fibres it innervates) are the fundamental elements of the neuromuscular system that transform motor commands into mechanical actions. The transformation of the activation signal from the nervous system into single muscle fibre potentials is a non-linear function comprising excitatory and inhibitory ionotropic synaptic inputs, which are subject to neuromodulation due to monoaminergic inputs (e.g. serotonin, noradrenaline). These diffuse neuromodulatory inputs augment motoneuron excitability by activating dendritic persistent inward currents that amplify and prolong the synaptic input. Whilst powerful in amplifying synaptic inputs, the diffuse nature of neuromodulatory inputs could also impair motor control by, for example, increasing excitability of the agonist and antagonist motor pools simultaneously. Persistent inward currents (PICs) are, however, highly sensitive to local inhibitory control. Diffuse facilitation of PICs via neuromodulatory input and their attenuation via local inhibitory circuits is thus required for maintaining normal motor control and function (Khurram et al. 2022). In humans, PICs cannot be measured directly, but the magnitude of PICs can be estimated by quantifying the onset–offset hysteresis of pairs of MUs that share common synaptic input, also known as the paired motor unit analysis (i.e. $\Delta F$) technique.

Ageing is associated with several alterations within the neuromuscular system that underlie the decline in muscle force production and control. For example, ageing adults typically exhibit lower MU discharge rates and compressed discharge rate modulation with increasing levels of voluntary effort. Recent evidence further suggests that the reduced discharge rate of MUs in older adults might be related to reduced PIC magnitudes (Hassan et al., 2021), though these findings are currently limited to motor pools of a few muscles (biceps and triceps brachii, soleus, tibialis anterior). Whilst the reduction in PIC magnitude has been attributed to the age-related deterioration within the monoaminergic system, ageing adults also exhibit deteriorations in local spinal circuitry. When investigating potential strategies to counteract the reduced PIC magnitude in older adults, the question is whether the ability to modulate this reduced PIC magnitude is also impaired. In other words, in addition to having reduced PIC magnitude, does our ability to facilitate or inhibit PICs also decrease as we get older? This question is addressed by Orssatto and colleagues (2022) in an article in this issue of *The Journal of Physiology*.

The approach taken by Orssatto and colleagues was to perform two experiments with different interventions designed to modulate estimates of PIC magnitude via two different mechanisms in young and older adults. In both experiments, PIC magnitude was estimated using the paired motor unit analysis (i.e. $\Delta F$) technique from MU population recordings in soleus and tibialis anterior during low-intensity (up to 20–30% of maximal voluntary force) triangular isometric contractions. In the first experiment, PIC magnitude was estimated before and after a remote sustained contraction (i.e. a handgrip), which is thought to augment the diffuse release of monoamines (i.e. serotonin) onto motoneurons and thus facilitate PICs. Indeed, they showed that $\Delta F$ increased with remote handgrip contraction and this increase was similar for both young and older adults, suggesting that the ability to facilitate PICs is maintained in older age. In the second experiment, PIC magnitude was estimated with and without vibration applied to the antagonist tendon. The vibration of the tendon excites Ia

afferent axons, which, via an interneuron, inhibits the antagonist motoneuron. Though elicited with electrical stimulation, reciprocal inhibition has been previously shown to be a powerful stimulus to reduce PIC magnitudes in younger adults (Mesquita et al., 2022), something which was confirmed by Orssatto and colleagues using a vibration stimulus. In older adults, however, antagonist vibration resulted in reduced PIC magnitude similar to young adults in soleus only, whereas the reciprocal inhibition appeared weaker in tibialis anterior.

Overall, the results of Orssatto and colleagues provide novel evidence in older adults that, despite their having reduced magnitude, the ability of PICs to modulate is possible via (a) increased diffuse monoaminergic drive, and (b) local inhibitory circuitry, though the latter may be limited in some muscles. These findings offer important insights into changes in motor control with ageing and provide a mechanistic basis for potential interventions to counteract the age-related decline in neuromuscular function. Moreover, they pave the way for further investigation into the nature of motor command changes with ageing. Firstly, and as discussed by the authors, the notion of a ceiling needs to be considered in future studies. The potential ceiling effect is relevant for both the facilitation and the inhibition of PICs, with older adults possibly having a greater and smaller potential for them, respectively, due to their lower PIC magnitudes. Secondly, the findings of Orssatto and colleagues are at present limited to motor pools controlling tibialis anterior and soleus, and thus investigation of the age-related changes in PIC magnitudes and the ability to modulate them needs to be expanded to other muscle groups. Thirdly, synaptic inputs received by motoneurons may vary according to their recruitment threshold (Khurram et al., 2022), and our current understanding of motor commands that shape MU discharge modulation in ageing adults is largely limited to lower contraction intensities and thus lower-threshold MUs, since current methodologies make the discrimination of discharge characteristics of higher-threshold MUs more difficult, though not impossible. Attempting to assess the modulation of discharge rates

in both lower- and higher-threshold MUs, in combination with novel computational methods for estimating the non-linearities in motor unit discharge introduced by PICs (Beauchamp et al., 2022) in addition to the paired motor unit analysis technique (i.e. $\Delta F$), will lead to an improved understanding of the age-related neuromuscular function decline and the design of strategies to counteract it. The study by Orssatto and colleagues is certainly one of the first steps on this exciting journey.

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

## Additional information

### Competing interests

None.

### Author contributions

Sole author.

## Funding

J.S. is supported by Versus Arthritis Foundation Fellowship (reference: 22569).

## Acknowledgments

The author thanks Dr Greg Pearcey, and my PhD students (Miss Tamara Valenčič, Mr Chris Connelly, and Mr Haydn Thomason) for a fruitful discussion in the process of writing this perspective.

## Keywords

ageing, monoamines, motor unit, motor control, neuromodulation

## Supporting information

Additional supporting information can be found online in the Supporting Information section at the end of the HTML view of the article. Supporting information files available:

**Peer Review History**

