## [Peer Review History · The Journal of Physiology]

Remote contractions to mitigate reduced persistent inward current magnitudes in motoneurons of older adults

Jakob Škarabot

DOI: 10.1113/JP283964

Corresponding author(s): Jakob Škarabot (J.Skarabot@lboro.ac.uk)

The following individual(s) involved in review of this submission have agreed to reveal their identity: Lucas B R Orssatto (Referee #1)

Review Timeline:

Submission Date:	24-Oct-2022
Editorial Decision:	28-Oct-2022
Revision Received:	28-Oct-2022
Accepted:	31-Oct-2022

Senior Editor: Richard Carson

Reviewing Editor: Mathew Piasecki

Transaction Report:

Dear Dr Škarabot,

Re: JP-P-2022-283964 "Remote contractions to mitigate reduced persistent inward current magnitudes in motoneurons of older adults" by Jakob Škarabot

Thank you for submitting your invited Perspectives article to The Journal of Physiology. It has been assessed by a Reviewing Editor and the author of the focus paper.

Minor alterations have been requested.

The reports are copied at the end of this email. Please address all of the points and incorporate all requested revisions.

NEW POLICY: In order to improve the transparency of its peer review process The Journal of Physiology publishes online as supporting information the peer review history of all articles accepted for publication. Readers will have access to decision letters, including all Editors' comments and referee reports, for each version of the manuscript and any author responses to peer review comments. Referees can decide whether or not they wish to be named on the peer review history document.

I hope you will find the comments helpful and have no difficulty in revising your article within 7 days.

To submit the revised version use the links in Author Tasks Link Not Available.

Please ensure that the article is a Word File with no more than 5 references, including the focus paper.

Thank you for your contribution to the Journal.

Yours sincerely,

Richard Carson
Senior Editor
The Journal of Physiology

EDITOR COMMENTS

Reviewing Editor:

This is a practical summary of the paper and the supporting literature. As you will see the reviewer has recommended highlighting the muscle specificity of these findings. Please address this where possible.

Please include a full title page as part of your article (Word) file (containing title, authors, affiliations, corresponding author name and contact details, keywords, and running title).

REFeree COMMENTS:

Referee #1:

We would like to commend Dr Skarabot for presenting a well-written and thoughtful perspective on the recent work by our group. The author initially summarised our work and subsequently proposed his viewpoint on the topic using a writing style accessible for a broad readership.

Our only suggestion would be to remind the reader that our findings were obtained from tibialis anterior and soleus only; therefore, extrapolation to other muscles should be done only with due caution, and this should be explored in future studies. For example, recent work by Mesquita et al. (2020) reported no changes in estimates of PICs in from gastrocnemius medialis motor units during jaw-clenching. And so far, it is only known that ΔF is reduced in older adults in triceps brachii, biceps brachii (Hassan et al., 2021), soleus, and tibialis anterior (Orssatto et al., 2021). Therefore, the behaviour described in our present study still needs to be explored across muscles.

Hassan, A. S., Fajardo, M. E., Cummings, M., McPherson, L. M., Negro, F., Dewald, J. P. A., Heckman, C. J., & Pearcey, G. E. (2021). Estimates of persistent inward currents are reduced in upper limb motor units of older adults. *The Journal of Physiology*. <https://doi.org/10.1113/jp282063>

Mesquita, R. N. O., Taylor, J. L., Trajano, G. S., Holobar, A., Gonçalves, B., & Blazeovich, A. J. (2020). Effect of Jaw-Clenching on Persistent Inward Currents. ISEK Virtual Congress Poster Abstract Booklet, 118-119. <https://isek.org/wp-content/uploads/2020/07/ISEK-Poster-Abstract-Booklet-Jul7.pdf>

Orsatto, L. B. R., Borg, D. N., Blazeovich, A. J., Sakugawa, R. L., Shield, A. J., & Trajano, G. S. (2021). Intrinsic motoneuron excitability is reduced in soleus and tibialis anterior of older adults. *GeroScience*, 43(6), 2719-2735. <https://doi.org/10.1007/s11357-021-00478-z>

Confidential Review

24-Oct-2022

We would like to commend Dr Skarabot for presenting a well-written and thoughtful perspective on the recent work by our group. The author initially summarised our work and subsequently proposed his viewpoint on the topic using a writing style accessible for a broad readership.

Our only suggestion would be to remind the reader that our findings were obtained from *tibialis anterior* and *soleus* only; therefore, extrapolation to other muscles should be done only with due caution, and this should be explored in future studies. For example, recent work by Mesquita et al. (2020) reported no changes in estimates of PICs in from *gastrocnemius medialis* motor units during jaw-clenching. And so far, it is only known that ΔF is reduced in older adults in triceps brachii, biceps brachii (Hassan et al., 2021), soleus, and tibialis anterior (Orssatto et al., 2021). Therefore, the behaviour described in our present study still needs to be explored across muscles.

Hassan, A. S., Fajardo, M. E., Cummings, M., McPherson, L. M., Negro, F., Dewald, J. P. A., Heckman, C. J., & Pearcey, G. E. (2021). Estimates of persistent inward currents are reduced in upper limb motor units of older adults. *The Journal of Physiology*. <https://doi.org/10.1113/jp282063>

Mesquita, R. N. O., Taylor, J. L., Trajano, G. S., Holobar, A., Gonçalves, B., & Blazeovich, A. J. (2020). Effect of Jaw-Clenching on Persistent Inward Currents. *ISEK Virtual Congress Poster Abstract Booklet*, 118–119. <https://isek.org/wp-content/uploads/2020/07/ISEK-Poster-Abstract-Booklet-Jul7.pdf>

Orssatto, L. B. R., Borg, D. N., Blazeovich, A. J., Sakugawa, R. L., Shield, A. J., & Trajano, G. S. (2021). Intrinsic motoneuron excitability is reduced in soleus and tibialis anterior of older adults. *GeroScience*, 43(6), 2719–2735. <https://doi.org/10.1007/s11357-021-00478-z>

Dear Dr Škarabot,

Re: JP-P-2022-283964 "Remote contractions to mitigate reduced persistent inward current magnitudes in motoneurons of older adults" by Jakob Škarabot

Thank you for submitting your invited Perspectives article to The Journal of Physiology. It has been assessed by a Reviewing Editor and the author of the focus paper.

Minor alterations have been requested.

The reports are copied at the end of this email. Please address all of the points and incorporate all requested revisions.

NEW POLICY: In order to improve the transparency of its peer review process The Journal of Physiology publishes online as supporting information the peer review history of all articles accepted for publication. Readers will have access to decision letters, including all Editors' comments and referee reports, for each version of the manuscript and any author responses to peer review comments. Referees can decide whether or not they wish to be named on the peer review history document.

I hope you will find the comments helpful and have no difficulty in revising your article within 7 days.

To submit the revised version use the links in Author Tasks <https://jp.msubmit.net/cgi-bin/main.plex?el=A6JS1FRd7A2SWp4F1A9ftdX6g0R1tzc2kE8JxWi4i1DAZ>.

Please ensure that the article is a Word File with no more than 5 references, including the focus paper.

Thank you for your contribution to the Journal.

Yours sincerely,

Richard Carson
Senior Editor
The Journal of Physiology

Thank you for the opportunity to improve the manuscript. I have addressed the comment raised by the authors of the original work.

EDITOR COMMENTS

Reviewing Editor:

This is a practical summary of the paper and the supporting literature. As you will see the reviewer has recommended highlighting the muscle specificity of these findings. Please address this where possible.

Please include a full title page as part of your article (Word) file (containing title, authors, affiliations, corresponding author name and contact details, keywords, and running title).

Thank you. I have addressed the comment raised by the authors. I have also now included the title page.

REFEREE COMMENTS:

Referee #1:

We would like to commend Dr Skarabot for presenting a well-written and thoughtful perspective on the recent work by our group. The author initially summarised our work and subsequently proposed his viewpoint on the topic using a writing style accessible for a broad readership.

I thank the authors for the kind words about my perspective and for providing a useful comment which I believe has improved the work.

Our only suggestion would be to remind the reader that our findings were obtained from tibialis anterior and soleus only; therefore, extrapolation to other muscles should be done only with due caution, and this should be explored in future studies. For example, recent work by Mesquita et al. (2020) reported no changes in estimates of PICs in from gastrocnemius medialis motor units during jaw-clenching. And so far, it is only known that ΔF is reduced in older adults in triceps brachii, biceps brachii (Hassan et al., 2021), soleus, and tibialis anterior (Orssatto et al., 2021). Therefore, the behaviour described in our present study still needs to be explored across muscles.

Hassan, A. S., Fajardo, M. E., Cummings, M., McPherson, L. M., Negro, F., Dewald, J. P. A., Heckman, C. J., & Pearcey, G. E. (2021). Estimates of persistent inward currents are reduced in upper limb motor units of older adults. *The Journal of Physiology*. <https://doi.org/10.1113/jp282063>

Mesquita, R. N. O., Taylor, J. L., Trajano, G. S., Holobar, A., Gonçalves, B., & Blazeovich, A. J. (2020). Effect of Jaw-Clenching on Persistent Inward Currents. ISEK Virtual Congress Poster Abstract Booklet, 118-119. <https://isek.org/wp-content/uploads/2020/07/ISEK-Poster-Abstract-Booklet-Jul7.pdf>

Orssatto, L. B. R., Borg, D. N., Blazeovich, A. J., Sakugawa, R. L., Shield, A. J., & Trajano, G. S. (2021). Intrinsic motoneuron excitability is reduced in soleus and tibialis anterior of older adults. *GeroScience*, 43(6), 2719-2735. <https://doi.org/10.1007/s11357-021-00478-z>

This is a great point. I have made some revisions that highlight that the age-related differences are presently limited to motor pools of only a few muscle groups, and reminded the reader that your findings are limited to soleus and tibialis anterior and should thus be explored in motor pools of other muscles/muscle groups.

Specifically, the sentence highlighting the differences in PIC magnitudes in young and older adults now reads: “Recent evidence further suggests that the reduced discharge rate of MUs in older adults might be related to reduced PIC magnitudes (Hassan et al., 2021), *though these findings are currently limited to motor pools of a few muscles (biceps and triceps brachii, soleus, tibialis anterior).*”

I have also added a sentence in the final paragraph: “*Secondly, the findings of Orssatto and colleagues are presently limited to motor pools controlling tibialis anterior and soleus, and thus investigation of the age-related changes in PIC magnitudes and the ability to modulate them needs to be expanded to other muscle groups.*”

The reference limit for this article type unfortunately precludes me from citing all the great references the authors have kindly provided.

Dear Dr Škarabot,

Re: JP-P-2022-283964R1 "Remote contractions to mitigate reduced persistent inward current magnitudes in motoneurons of older adults" by Jakob Škarabot

I am pleased to tell you that your invited Perspective article has been accepted for publication in The Journal of Physiology.

NEW POLICY: In order to improve the transparency of its peer review process, The Journal of Physiology publishes online as supporting information the peer review history of all articles accepted for publication. Readers will have access to decision letters, including all Editors' comments and referee reports, for each version of the manuscript and any author responses to peer review comments. Referees can decide whether or not they wish to be named on the peer review history document.

The last Word version of the paper submitted will be used by the Production Editors to prepare your proof. When this is ready you will receive an email containing a link to Wiley's Online Proofing System. The proof should be checked and corrected as quickly as possible.

All queries at proof stage should be sent to tjp@wiley.com.

Thank you very much for your contribution to The Journal of Physiology.

Yours sincerely,

Richard Carson
Senior Editor
The Journal of Physiology

Reviewing Editor Comments:

Thank you for addressing the additional comments.